# Morphological Features and Cold-Response Gene Expression in Mesophilic *Bacillus cereus* Group and Psychrotolerant *Bacillus cereus* Group under Low Temperature

**DOI:** 10.3390/microorganisms9061255

**Published:** 2021-06-09

**Authors:** Kyung-Min Park, Hyun-Jung Kim, Min-Sun Kim, Minseon Koo

**Affiliations:** 1Department of Food Analysis Center, Korea Research Institute, Wanju-gun 55365, Jeollabuk-do, Korea; Parkkyungmin@kfri.re.kr (K.-M.P.); mskim@kfri.re.kr (M.-S.K.); 2Research Group of Consumer Safety, Korea Food Research Institute, Wanju-gun 55365, Jeollabuk-do, Korea; hjkim@kfri.re.kr; 3Department of Food Biotechnology, Korea University of Science & Technology, Daejeon 34113, Korea

**Keywords:** *Bacillus cereus* group, morphology, transcriptome, low temperature, peptidoglycan biosynthesis, energy conservation

## Abstract

At low temperatures, psychrotolerant *B. cereus* group strains exhibit a higher growth rate than mesophilic strains do. However, the different survival responses of the psychrotolerant strain (BCG^34^) and the mesophilic strain (BCG^T^) at low temperatures are unclear. We investigated the morphological and genomic features of BCG^T^ and BCG^34^ to characterize their growth strategies at low temperatures. At low temperatures, morphological changes were observed only in BCG^T^. These morphological changes included the elongation of rod-shaped cells, whereas the cell shape in BCG^34^ was unchanged at the low temperature. A transcriptomic analysis revealed that both species exhibited different growth-related traits during low-temperature growth. The BCG^T^ strain induces fatty acid biosynthesis, sulfur assimilation, and methionine and cysteine biosynthesis as a survival mechanism in cold systems. Increases in energy metabolism and fatty acid biosynthesis in the mesophilic *B. cereus* group strain might explain its ability to grow at low temperatures. Several pathways involved in carbohydrate mechanisms were downregulated to conserve the energy required for growth. Peptidoglycan biosynthesis was upregulated, implying that a change of gene expression in both RNA-Seq and RT-qPCR contributed to sustaining its growth and rod shape at low temperatures. These results improve our understanding of the growth response of the *B. cereus* group, including psychrotolerant *B. cereus* group strains, at low temperatures and provide information for improving bacterial inhibition strategies in the food industry.

## 1. Introduction

The Bacillus cereus group comprises more than 20 species with close genetic similarity and nine species, namely, *B. anthracis*, *B. cereus*, *B. cytotoxicus*, *B. mycoides*, *B. pseudomycoides*, *B. thuringiensis*, *B. toyonensis*, *B. weihenstephanensis*, and *B. wiedmannii* that can cause anthrax or foodborne illness in humans or insects [1,2,3] depending on the presence and expression of virulence genes. Psychrotolerant *B. cereus* group strains, such as *B. weihenstephanensis* and *B. wiedmannii*, can grow at 7 °C or below, whereas other *B. cereus* group strains cannot grow at less than 10 °C [3,4,5]. The psychrotolerant *B. cereus* strains can survive during the production, distribution, and storage of various refrigerated foods, such as dairy products [6]. As the psychrotolerant *B. cereus* group strains possess enterotoxin or emetic toxin genes [3,7], their existence in refrigerated food products should be monitored, and the control of these species is crucial for improving food safety.

Consumer demand for refrigerated foods has increased due to the retention of nutritional and sensorial qualities in foodstuffs. Cold storage might provide an environment that is favorable for bacterial survival and the growth of psychrotolerant *B. cereus* strains. A temperature lower than the optimal temperature poses many challenges to bacteria. Exposure to low temperature increases the fluidity of the cell membrane by modifying the fatty acid composition, such as the unsaturated fatty acid composition [8]. Bacteria repress protein-folding and stabilize nucleic acid structures by increasing RNA transcription and translation to allow for their survival at low temperatures [9]. The impact of low temperature on the survival and growth of bacteria has been reported. According to Becker et al. [10], *Listeria monocytogenes* utilizes an adaptive response mechanism under cold stress that involves a modification of the fatty acid composition of its bacterial membrane to maintain the fluidity of the membrane. The adaptation of *Escherichia coli* to low temperatures is associated with an increasing ratio of unsaturated fatty acids to saturated fatty acids [11]. In *Bacillus subtilis*, the stress response to low temperatures involves a high content of branched fatty acids and proteins involved in the translation machinery [12]. The growth of *B. cereus* at low temperatures is also associated with a higher proportion of unsaturated fatty acids in the bacterial membrane than that found at optimal growth temperatures [13]. In general, bacteria regulate cellular factors such as fatty acid desaturases [13], cold-shock proteins [14], and transcriptional regulators [15,16] to survive at low temperatures.

*B. cereus* group is not traditionally considered a psychrotrophic species, but psychrotolerant *B. cereus* group strains possess a growth ability at below 10 °C and have successfully colonized at low temperatures without cold adaptation. In a previous study, we isolated a psychrotolerant *B. cereus* group strain from fresh vegetables distributed by cold supply chains. Some isolates were grown at less than 10 °C and showed greater biofilm formation at low temperatures than at an optimal temperature [17]. The survival response in cold adaptation strategies of *B. cereus* under cold conditions has been studied, but limited information is available on the growth response of psychrotolerant *B. cereus* group strains (which are able to colonize without cold adaptation) under cold conditions. In this study, we investigated whether a psychrotolerant *B. cereus* group strain reveals a similar response to growth at low temperatures as a mesophilic *B. cereus* group strain. This study compared the relevant transcriptomic changes and morphological changes in each *B. cereus* group strain using field-emission scanning electron microscopy (FE-SEM) and RNA sequencing at optimal (30 °C) and low temperatures (10 °C) to confirm the morphological and transcriptomic changes associated with the growth of a mesophilic *B. cereus* strain and a psychrotolerant *B. cereus* group strain at low temperature. The present study may provide a better understanding of the bacterial growth response employed by *B. cereus* group strains, including psychrotolerant strains, at low temperatures.

## 2. Materials and Methods

### 2.1. Bacterial Strains and Growth Conditions

The experiments conducted in this study were performed with a mesophilic *B. cereus* strain (*B. cereus* ATCC 14579, BCG^T^) and a psychrotolerant *B. cereus* strain (food isolate, BCG^34^). The food isolate BCG34 is a psychrotolerant strain isolated from our previous study [17]. The two types of *B. cereus* group were cultured using tryptic soy broth (TSB, Merck, Darmstadt, Germany) at 30 °C. As the mesophilic *B. cereus* group (BCG^T^) did not grow without the adaptation process at the low temperature, pre-cultures were propagated to an OD_600_ of 0.5 at 30 °C, diluted to an OD_600_ of 0.1, and subsequently transferred to a lower growth temperature (20 °C) depending on the individual experiments for the adaptation of BCG^T^ to low temperatures. The growth regimen applied allowed the growth of the cultures to an OD_600_ of at least 6, and the cultures were propagated at a lower temperature (15 °C) for at least three generations before harvesting. Fifteen days after the transfer to 10 °C, when the OD_600_ exceeded 0.6, the cultures were harvested for growth capability evaluation and RNA extraction. The psychrotolerant strain BCG^34^ was activated at 30 °C, transferred to 10 °C, and incubated for 8 days (OD_600_>0.6) without cold adaptation processing. BCG^T^ and BCG^34^ grown at 30 °C were included as controls for RNA sequencing.

### 2.2. Bacterial Growth Capability at Low Temperature

The mesophilic (BCG^T^) *B. cereus* strain and the psychrotolerant (BCG^34^) *B. cereus* strain were first incubated overnight at 30 °C. The overnight cultures of BCG^T^ and BCG^34^ were activated at 10 °C, diluted to a final concentration of 1 × 10^3^ log CFU/mL, and then incubated in triplicate at 10 °C for 12 days and 7 °C for 30 days to evaluate the growth capability at low temperatures. The bacterial suspension was cultured at various time intervals (0, 5, 10, 15, 20, 25, and 30 days for 7 °C and 0, 2, 4, 6, 8, 10, and 12 days for 10 °C) during growth at low temperatures, from which aliquots were serially diluted in 0.85% sterile saline. Then, 100 µL of each dilution was inoculated on Mannitol egg yolk polymyxin agar (MYP, Merck, Germany) in duplicate, followed by incubation at 30 °C for 24 h before a colony count was performed.

### 2.3. Field-Emission Scanning Electron Microscopy (FE-SEM)

To obtain vegetative cells (endospore-free suspension), the mesophilic (BCG^T^) *B. cereus* group strain and the psychrotolerant (BCG^34^) *B. cereus* group strain were first incubated overnight at 30 °C, and 1 mL suspensions were then poured in 99 mL of TSB. The final suspensions were allowed to grow at 30 °C for 4 h, when the cell density reached 7 log CFU/mL [18]. In addition, BCG^T^ and BCG^34^ also grew at 10 °C for 10 days (for BCG^T^) and 8 days (for BCG^34^) to collect the vegetative cells activated at low temperatures. For the preprocessing of the samples for field-emission scanning electron microscopy (FE-SEM), the cells activated at 30 and 10 °C were collected by centrifugation at 2320× *g* and 4 °C for 10 min, washed twice with 0.85% sterile saline, fixed with 2.0% paraformaldehyde and 2.0% glutaraldehyde in 0.05 M cacodylate buffer (pH 7.2) overnight at 4°C, and washed three times with 0.05 M sodium cacodylate buffer (pH 7.2) at 2320× *g* for 5 min. The samples were then postfixed with 1% osmium tetroxide in 0.05 M cacodylate buffer (pH 7.2) for 1.5 h at 4 °C and again washed three times with distilled water. Subsequently, the specimens were dehydrated through a graded ethanol series (30, 40, 50, 70, 80, 90, and 100%). The drying process was conducted twice with 100% hexamethyldisilazane (HMDS) for 5 min. Afterward, a platinum film coating made the cell surface observable by an FE-SEM instrument (SUPRA 40 VP, Carl Zeiss Oberkochen, Germany).

### 2.4. RNA Sequencing

#### 2.4.1. Cell Lysis and RNA Isolation

Briefly, total RNA from the mesophilic (BCG^T^) *B. cereus* group strain and the psychrotolerant (BCG^34^) *B. cereus* group strain grown at 30 and 10 °C in TSB broth to the mid-exponential phase (OD_600_ > 0.6) was extracted using an RNA plus mini kit (Qiagen, Germany) according to the manufacturer’s protocol for next-generation sequencing and quantitative real-time PCR (RT-qPCR) assay. Duplicates of RNA samples with an RNA integrity number > 7.0 were used for RNA sequencing by Bioneer Company (Daejeon, Korea). The purity, concentration, and integrity of the RNA samples were determined using NanoDrop (Thermo Scientific, Wilmington, DE, USA), Agilent 2100, and Qubit 2.0 instruments, respectively. The RNA samples were purified and fragmented using the TruSeq RNA Sample Preparation Kit v2-Set A (Illumina, San Diego, CA, USA) for library construction.

#### 2.4.2. Illumina Sequencing

*B. cereus* transcriptomes were sequenced on the Illumina HiSeq 2500 (Illumina, San Diego, CA, USA) platform, and de novo assembly was performed with the short paired end reads. Prior to transcript assembly, all raw reads were processed to remove low-quality and adapter sequences. After filtering, the trimmed reads were assessed using FastQC (https://www.bioinformatics.babraham.ac.uk/projects/fastqc/ (accessed on 16 January 2020)). The preprocessed reads were aligned to the *B. cereus* group reference genome (GenBank Accessions: NZ_CP034551, NZ_CP064875, NZ_CP021061, NZ_CP032365, and NZ_CP000903) to obtain high-quality clean reads. Finally, the reads from BCG^T^ and BCG^34^ were mapped to the complete sequenced genomes of NZ_CP034551 and NZ_CP000903, respectively. The clean reads were then assembled into contigs, transcripts, and unigenes using the Trinity platform. The raw data have been submitted to the NCBI Sequence Read Archive with the accession number PRJNA727406.

#### 2.4.3. Differential Gene Expression Analysis

Bowtie was used to align the read reference sequences [19], and the expression levels were estimated by RSEM [20]. The gene expression level was measured as the number of clean reads mapped to the reference sequence and normalized to fragments per kilobase per million (FPKM) fragments. Differential expression analysis was conducted via pairwise comparisons using the Benjamini–Hochberg method with EBSeq software [21]. The false discovery rate (FDR) was used to determine the threshold P-value in multiple tests and analyses. The criteria FDR < 0.01 and absolute value of log_2_ (FC) ≥ 1 were used as thresholds to define significantly different gene expression levels [22]. Functional classification of the DEGs was conducted through BLAST analysis as described above. GO enrichment analysis and KEGG pathway enrichment analysis were performed using DAVID bioinformatics resources 6.8 (https://david.ncifcrf.gov/ (accessed on 16 January 2020)). For pathway enrichment analysis, all DEGs were aligned to terms in the KEGG pathway database (https://www.genome.jp/kegg/pathway.html (accessed on 16 January 2016)) and searched to identify significantly enriched KEGG terms.

### 2.5. Gene Expression Analysis Using Reverse Transcription Quantitative PCR (RT-qPCR)

Total RNA was extracted using an RNA plus mini kit (Qiagen, Hilden, Germany) from the mesophilic *B. cereus* group strain and the psychrotolerant *B. cereus* group strain. Complementary DNA was synthesized using a High-Capacity RNA-to-cDNA^TM^ kit (Thermo Fisher Scientific, Waltham, MA, USA) following the manufacturer’s instructions. Three independent experiments were performed. Six differentially expressed genes (DEGs) with a twofold change and a *p*-value < 0.05 were selected for RT-qPCR analysis. The qPCR was performed using 20 uL reaction volume with a cDNA template, SYBR premix, and primer pairs by ABI QuantStudio™ 3 real time system (ABI Systems, Foster City, CA, USA). The housekeeping gene *rpoB* was used as the reference gene to normalize and estimate the up- or downregulation of the selected genes. Gene-specific primer pairs were designed using the NCBI primer BLAST tool (https://www.ncbi.nlm.nih.gov/tools/primer-blast (accessed on 9 April 2021)) and Primer3Plus (https://www.bioinformatics.nl/cgi-bin/primer3plus/primer3plus.cgi (accessed on 9 April 2021)). All primers are listed in Table 1. The relative expression level of each gene was calculated using the 2^−∆∆Ct^ method with a reference gene. The results were expressed as the fold change (log_2_) between 30 and 10 °C.

The thermal cycling protocol was as follows: initial denaturation for 10 min at 95 °C followed by 40 cycles of 15 s at 95 °C, 30 s at 60 °C, and 30 s at 72 °C. The fluorescence signal was measured at the end of each extension step at 72 °C. After the amplification, a melting curve analysis with a temperature gradient of 0.1 °C/s from 60 to 95 °C was performed to confirm that only the specific products were amplified. Finally, the samples were cooled to 20 °C for 10 s. The 2^−ΔΔCt^ method was used to evaluate the expression levels of each target gene compared with the housekeeping gene, *rpoB*. The results were expressed as the fold change (log_2_) between 30 and 10 °C, and the results of the RNA qPCR were consistent with RNA-Seq data. All the experiments were performed in duplicate, and the data presented were obtained from at least three independent experiments.

### 2.6. Statistical Analyses

Statistical analyses were performed using Microsoft Excel 2018. The growth ability test for the mesophilic *B. cereus* group strain and the psychrotolerant *B. cereus* group strain was carried out in triplicate, and the mean values and standard deviations were calculated.

## 3. Results

### 3.1. Growth Capability of Mesophilic B. cereus Group Strain and Psychrotolerant B. cereus Group Strain Activated at Low Temperature

BCG^T^ and BCG^34^ were selected as the mesophilic *B. cereus* group strain and the psychrotolerant *B. cereus* group strain, respectively, and the growth rate of these strains at low temperatures was tested. Although BCG^T^ and BCG^34^ grew at 10 °C, BCG^34^ showed a higher growth rate than BCG^T^ did, with a lag time of 0.657 and a specific maximum rate of 1.3 (Figure 1a,b).

At 7 °C, BCG^34^ exhibited a reduced ability to grow, whereas BCG^T^ was completely unable to grow. Therefore, the morphological and transcriptomic changes in the mesophilic *B. cereus* group strain and the psychrotolerant *B. cereus* group strain at low temperatures were compared to understand the differences in the bacterial survival responses between these two different *B. cereus* groups.

### 3.2. Microscopic Observations of Mesophilic B. cereus Group Strain and Psychrotolerant B. cereus Group Strain at Low Temperature

The morphological changes in the mesophilic *B. cereus* group strain and the psychrotolerant *B. cereus* group strain after cultivation at optimal and low temperatures were observed by FE-SEM, and these observations provide a better understanding of the growth abilities of these different *B. cereus* groups at low temperatures. After cultivation at 30 °C, BCG^T^ and BCG^34^ showed similar features of intact and striated membranes with rough surfaces (Figure 2a,d). BCG^T^ and BCG^34^ cells grown at 30 °C also exhibited a normal rod morphology with a cell length of 1934~2753 µm, but BCG^T^ and BCG^34^ cells cultivated at 10 °C showed different morphological features (Figure 2b,c for BCG^T^ and Figure 2e,f for BCG^34^). Specifically, BCG^T^ cells grown at 10 °C were more elongated than those grown at 30 °C, and the cell lengths detected in the experiment ranged from an average of 2.742 µm (observed for BCG^34^ cells) to a maximum of 7890 µm (observed for BCG^T^ cells). The formation of elongated cells under cold conditions was observed only in BCG^T^ cells. These results are consistent with those obtained by other researchers. Den Besten et al. [23] observed individual *B. cereus* cells forming filaments during growth under salt stress conditions. Aerobic cultures of *B. cereus* ATCC 14579 maintained at 15 °C produced larger cells than those obtained after culture at 37 °C [24].

Cell division occurs in different stages: DNA duplication, lengthening of cells, septum formation, septum finalization, and the separation of cells [25]. Elongation and division require the biosynthesis of peptidoglycan, which is an essential component of the cell wall that protects bacteria from environmental stress. The carefully coordinated biosynthesis of peptidoglycans during cell elongation and division is needed for cell viability [26]. The inhibition of cell wall synthesis by the disruption of homeostasis between elongation and division might lead to bacterial death. At low temperatures, some BCG^T^ cells formed pores, whereas others were markedly shrunken and collapsed, and some cell walls and membranes were partially disrupted (Figure 2b). However, the majority of the BCG^34^ cells activated at 10 °C maintained their original shape, as revealed by the observation that most of the cell wall and cell membrane remained intact, and this maintenance of the cell wall integrity resulted in little outflow of the cellular contents, such as proteins and nucleic acids. Some researchers have reported that the mechanisms of antimicrobial agents include cell wall degradation, damage to the cytoplasmic membrane or membrane proteins, leakage of intracellular contents, coagulation of the cytoplasm, and/or depletion of the proton motive force [27]. When activated at low temperatures, BCG^T^ cells may be unable to maintain their proton motive force, and an imbalance of these forces induces a morphological change that leads to the degradation of peptidoglycan and ultimately results in the disruption of the turgor pressure of the cell membrane as well as the release of internal and cell-wall-associated enzymes [28]. The difference in morphological changes between the mesophilic *B. cereus* group strain and the psychrotolerant *B. cereus* group strain cultivated at 10 °C suggests that cold-activated mesophilic *B. cereus* group isolates may be more susceptible to other environmental stress conditions than the psychrotolerant *B. cereus* group isolates activated at 10 °C.

### 3.3. Differential Expression Analysis of Cold-Response-Related Genes

The results from the analysis of the RNA database of the mesophilic *B. cereus* group strain and the psychrotolerant *B. cereus* group strain at low temperatures are shown in Table 2.

A total of 28,171,092 reads at 30 °C (BCG^T^ _C) and 30,479,776 reads at 10 °C (BCG^T^ _T) were identified from the mesophilic *B. cereus* group strain. A total of 25,381,698 reads at 30 °C (BCG^34^_C) and 36,039,900 reads at 10 °C (BCG^34^_T) were identified from the psychrotolerant *B. cereus* group strain. To investigate the impact of low temperature on the survival of the mesophilic (BCG^T^) *B. cereus* group strain, a DEG analysis was performed to compare 30 °C and 10 °C among the biological replicates obtained under each condition. A total of 317 genes were differentially expressed, and among these, 172 and 145 genes showed increased and decreased expression, respectively, in BCG^T^ (Figure 3). 

According to the gene ontology (GO) classifications, the transcripts in BCG^T^ that showed differential expression between 30 °C and 10 °C were enriched in the small molecule biosynthetic process, organic acid biosynthetic process, lipid metabolic process, and carboxylic acid biosynthetic process (Figure 4a). However, a total of 202 genes were differentially expressed, and 85 and 117 genes showed increased and decreased expression, respectively, in BCG^34^ (Figure 3). Most DEGs in BCG^34^ were classified into the subcategories biosynthetic process, response to low temperature, tRNA-aminoacylation, nitrogen compound metabolism process, cellular protein metabolism process, peptide metabolic process, and translation within the biological process category (Figure 4b). Transcripts involved in the glycolysis pathway, tricarboxylic acid (TCA) cycle, ribosome pathway, antibiotic biosynthesis pathway, and secondary metabolite biosynthesis pathways were downregulated at 10 °C compared with 30 °C, and those involved in peptidoglycan biosynthesis and cell wall biosynthesis were upregulated.

#### 3.3.1. Lipid Metabolism

In bacteria, fatty acid metabolism is tightly regulated to allow a rapid response to changes in the environment. Moreover, the fatty acid composition of the cell membrane varies depending on the environmental conditions because it plays a leading role in bacterial adaptation to environmental changes [24]. Under cold stress, increases in the unsaturated fatty acid components in the bacterial membrane constitute an adaptive and survival mechanism to increase the flexibility of cold-adapted bacteria [29]. In the BCG^T^ strain, *fabG* (Log_2_FC: 4.61) and *fabH2* (Log_2_FC: 3.09), which are related to fatty acid biosynthesis, were upregulated by over 3.0-fold, and *fabZ*, which is involved in the first stage of membrane lipid biogenesis, was upregulated by 2.85-fold (Table 3). The genes *lcfA, fadE*, and *fadR*, which are related to fatty acid degradation, showed significantly increased expression by 3.08-, 4.63-, and 6.84-fold, respectively.

However, the expression levels of *fabG* and *fabH2* were decreased by 0.57- and 0.83-fold, respectively, in the psychrotolerant *B. cereus* group strain at 10 °C in comparison with their levels at 30 °C. The *fadE* gene related to fatty acid degradation was upregulated by 3.05-fold. RNA-Seq analysis revealed that the *fabG, fabH*, and *fabZ* genes related to fatty acid biosynthesis commonly showed significantly increased expression in BCG^T^ at low temperatures. As these genes are essential for fatty acid biosynthesis, their increased expression in the mesophilic *B. cereus* group at low temperatures might indicate the availability and utilization of fatty acids for bacterial growth. Furthermore, the observed upregulation of *fadR*, which is involved in blocking fatty acid oxidation by repressing *lcfA, fadR, fadB, fadA, fadE, fadH*, and *fadG*, which are related to fatty acid degradation, indicates that the inhibition of fatty acid degradation is needed to prevent membrane lysis and maintain the cell wall composition at low temperatures. Fatty acid biosynthesis was negatively regulated in the psychrotolerant *B.* cereus group strain at 10 °C, and the downregulation of genes related to fatty acid metabolism may be associated with repression of catabolic metabolism for providing an energy source. This result suggests a mechanism of energy-saving in the psychrotolerant *B. cereus* group strain for bacterial growth at low temperatures.

#### 3.3.2. Energy Metabolism

The expression of nine genes involved in sulfur metabolism, namely *sat*, CYSC1_BACHD, *cysH, ssuA, cysA2, cysW, cysA, fbpB*, and *cysU*, in the BCG^T^ strain at 10 °C was higher than that at 30 °C (Table 4).

These genes play an important role in sulfur assimilation metabolism for the biosynthesis of L-cysteine and methionine. For all microorganisms, sulfur, in addition to carbon and nitrogen, is an essential element for growth, although its elemental amount in living organisms is generally very low [30]. Sulfur is needed primarily as a component of the amino acids cysteine and methionine; plays an essential role in a variety of enzyme cofactors, including biotin, coenzyme A, coenzyme M, thiamine, and lipoic acid; and is critical in many redox processes [31]. Cysteine and sulfate are the preferred sources of sulfur for bacterial species. Moreover, cysteine-derived proteins such as thioredoxin play a central role in protection against oxidative stress. Sulfur for biosynthetic processes is derived from the assimilation of inorganic sulfate by bacteria. Sulfate is taken up by cells from the ambient environment via the ATP-dependent transporter complex CysUWA [32]. CysU and CysW are subunits that form channels in the inner membrane, and CysA is the predicted ATPase subunit driving sulfate translocation [33]. The *ssuBACE* gene cluster is needed for the utilization of aliphatic sulfonates as sulfur sources [34,35], and the *cysH* and *sat* genes encode sulfate transporters [36].

In this study, the BCG^T^ strain activated at low temperatures encoded the set of genes responsible for sulfur metabolism, including the *cysC* (CYSC1_BACHD), *cysH*, and *sat* genes, which are involved in assimilated sulfate reduction. In addition, the BCG^T^ strain possessed a sulfonate transport system that included *ssuA* and a sulfate transport system that included *cysWUA*. Furthermore, *metE* and OAH sulfhydrylase, which are related to cysteine biosynthesis in the BCG^T^ strain, were found at higher levels at 10 °C than at 30 °C. These results showed a trend similar to that reported by Huillet et al. [37], who demonstrated the existence of regulatory connections between sulfur metabolism for cysteine biosynthesis and the oxidative stress response in a mesophilic *B. cereus* group strain. Our results suggest that increases in genes related to sulfur assimilation metabolism and cysteine biosynthesis might ensure bacterial survival during the growth of mesophilic *B. cereus* group strains at low temperatures. However, the genes associated with sulfur metabolism were downregulated in the psychrotolerant *B. cereus* group strain, and the expression of these genes did not show significant differences between the two temperatures.

#### 3.3.3. Amino Acid Metabolism

Arginine is an important nutritional source during bacterial growth and serves as an important energy resource in bacteria by participating in the arginine deiminase (ADI) and arginase pathways. Arginine metabolism via the ADI pathway is widely present in a variety of Gram-positive bacteria, including *Bacillus* spp., *L. monocytogenes*, and several lactic acid bacteria, and enables the adaptation to hostile environmental niches and host defenses [38]. The ADI pathway is a multienzyme pathway encoded by arc operon genes, namely *arcA, arcB*, and *arcC*, which hydrolyze arginine to ornithine to form ammonia, CO_2_, and ATP [39]. When *B. cereus* is exposed to pH 5.5, *arcA*-*arcB*-*arcC* are significantly upregulated in acid-adapted cells compared with non-adapted cells [40]. This result is not consistent with our results. The *arcA*-*arcB*-*arcC* genes of the arginine catabolic operon in the BCG^T^ strain activated at a low temperature were downregulated 6- to 9-fold at 10 °C compared with the levels at 30 °C, and the *argH*-, *argF*-, and *argJ*-encoded enzymes needed for the conversion of glutamate to arginine (arginine biosynthesis) were also downregulated (3- to 7-fold) (Table 5). The downregulation of genes related to arginine catabolism might induce decreases in arginine and citrulline and the accumulation of ornithine. Ornithine is the intermediate compound in arginine biosynthesis and can act as a precursor of proline, which is known to serve as an osmoprotective compound under stress conditions [41,42]. In the BCG^T^ strain activated at a low temperature, the downregulation of the *arcA, arcB, argF*, and *argH* genes may be able to increase ornithine accumulation and may be associated with bacterial growth of the mesophilic *B. cereus* group strain under cold stress.

The changes in the expression of genes related to arginine metabolism at the low temperature did not significantly differ between BCG^34^ and BCG^T^. However, two genes (*kynU* and *kynB*) related to the production of nicotinamide adenine dinucleotide (NAD^+^) from tryptophan catabolism were increased. NAD^+^ is utilized as an important cofactor in microbial cellular processes.

#### 3.3.4. Carbohydrate Metabolism

The expression of genes involved in glycolysis, the TCA cycle, butanoate metabolism, and pentose phosphate metabolism was downregulated in the mesophilic *B. cereus* groupstrain and the psychrotolerant *B. cereus* group strain at 10 °C compared with that at 30 °C. As shown in Table 6, seven genes (*acoC, ldh2, acoL, eno, fba, pdhA*, and *ldh3*) associated with key glycolysis enzymes and three genes (*odhA, pdhD*, and *odhB*) involved in the TCA cycle were downregulated in the BCG^T^ strain at 10 °C compared with the levels at 30 °C, which suggests the repression of glycolysis and the TCA cycle at low temperatures. An early event observed during exposure to stress is the downregulation of energy and amino acid metabolism, which might constitute an energy conservation strategy and could represent the transition from bacterial growth to the induction of protective mechanisms [43]. A previous report revealed that stress conditions during bacterial growth decrease cellular respiration and downregulate glycolysis, pyruvate metabolism, and the TCA cycle [44].

An analysis of the expression of genes related to the central metabolic pathway (glycolysis) in BCG^34^ revealed 3.8- to 8.1-fold decreases in fructose-bisphosphate aldolase (*fba*, log_2_FC of −3.89), 2,6-dichlorophenolindophenol oxidoreductase subunit alpha and beta (*acoA*, log_2_FC of −5.82; *acoB*, log_2_FC of −6.29; *acoC*, log_2_FC of −5.99), L-lactate dehydrogenase 2 (*ldh2*, log_2_FC of −6.63), and pyruvate formate-lyase-activating enzyme (*pflA*, log_2_FC of −8.17).

Four genes (*phdA, phdB, odhA*, and *odhB*) involved in the TCA cycle were also decreased in BCG^34^ at 10 °C compared with their levels at 30 °C, whereas the *cstA* (carbon starvation protein A) and *adhB* (alcohol dehydrogenase 2) genes were upregulated. Alcohol dehydrogenase (*adh*) is an NAD(P)-dependent dehydrogenase to which the maintenance of energy metabolism and ATP levels needed for bacterial growth has been ascribed [45]. In *E. coli*, an increase in carbon-starvation-induced protein (*cstA*) helps bacteria to survive under carbon- and energy-source-limited conditions during exposure to stress [46].

The metabolic changes experienced by bacteria during growth under low-temperature conditions generally involve the repression of glycolysis and the TCA cycle [47], which is in accordance with our results. During growth at the low temperature, the psychrotolerant strain seemed to prefer to save energy, as indicated by the finding that several genes involved in glycolysis, TCA cycle, pentose phosphate cycle, and butanoate metabolism were downregulated at the low temperature compared to cells activated at the optimal temperature. The downshift of carbohydrate-related metabolic pathways shown in the psychrotolerant *B. cereus* group strain suggests that catabolic response was inhibited to reserve energy and maintain the growth ability at low temperatures.

#### 3.3.5. Cell Wall/Membrane

*Bacillus* spp. growing under stress conditions exhibit alterations in their cell wall or membrane [48]. During growth at the low temperature, the *epsC* gene of the category “cell wall” was downregulated in BCG^T^ relative to its expression at 30 °C, and the *murG1* and *uppP1* genes involved in peptidoglycan biosynthesis were also repressed (Table 7).

The genes involved in peptidoglycan biosynthesis are essential for the elongation and division of many rod-shaped bacteria and are associated with cell wall synthesis [49,50]. Therefore, the observed repression of these genes’ (*murG1, uppP1*, and *epsC*) expression was found to be related to peptidoglycan or cell wall biosynthesis.

However, the *flgB* gene, which encodes the flagellar basal body rod protein, was upregulated in BCG^T^ at 10 °C compared with its expression at 30 °C (Table 7). During exposure to environmental stress, chemotaxis and the ability to swim are important features for bacterial survival. Furthermore, the cell length of *B. subtilis* generally elongates with increases in flagellar basal bodies [51,52,53]. The temperature-dependent induction of flagellum synthesis has been observed in various bacteria, including pathogenic *L. monocytogenes* [54], in which Csp proteins regulate flagellum gene expression [55]. In addition, *Yersinia enterocolitica* [56], which is a cold-tolerant bacterium, overexpresses genes related to flagellar biosynthesis [57]. However, in BCG^34^, the genes related to peptidoglycan and cell wall biosynthesis were increased at the low temperature compared with the levels found at the optimal temperature; specifically, a 1.2- to 2.7-fold upregulation of three genes, *murG1* (2.21 log_2_FC), *uppP1* (1.27 log_2_FC), and *epsC* (2.67 log_2_FC), was observed. These results are in accordance with previous results. Raymond-Bouchard et al. reported that the cold-adapted bacteria *Polaromonas* sp. upregulates genes such as undecaprenyl diphosphate synthase and UDP-N-acetylmuramyl-tripeptide synthase (*murE*) for peptidoglycan biosynthesis [58]. Mykytczuk et al. [45] found that *Planococcus halocryophilus* increases its expression of genes responsible for peptidoglycan synthesis during growth at low temperatures. Since the cell wall/membrane plays an important role in bacterial growth under changed environmental conditions, the upregulation of genes related to peptidoglycan and cell wall biosynthesis in the psychrotolerant *B. cereus* strain may be used as a growth response to maintain the bacterial cell proliferation and cell rod shape at low temperatures.

## 4. Conclusions

Storage at low temperature is an effective method for maintaining food quality and inhibiting the growth of foodborne pathogens. However, this strategy might not be appropriate for inhibiting the growth of psychrotolerant *B. cereus* group strains because these strains are able to survive with a high growth rate under cold conditions. Understanding of the mechanisms associated with bacterial growth at low temperatures is needed to improve the systems used for the management of food safety by inhibiting the frequency of *B. cereus* group proliferation in foodstuffs distributed via cold supply chains. Our transcriptomic analysis of a mesophilic *B. cereus* group strain and a psychrotolerant *B. cereus* group strain grown at low temperatures revealed that these strains utilize different strategies for survival under cold conditions. The mesophilic *B. cereus* group strain used common bacterial stress mechanisms to grow at the low temperature. For example, this strain exhibited increasing sulfur assimilation, induction of cysteine and glutathione biosynthesis, and the upregulation of fatty acid biosynthesis for bacterial growth under cold conditions. Meanwhile, carbohydrate metabolism in the psychrotolerant *B. cereus* group strain was repressed to save energy for growth, while the upregulation of the genes responsible for the catabolism of tryptophan and alcohols was modulated for energy production by providing alternate pathways for NAD regeneration and the balance of ATP levels. Furthermore, increased peptidoglycan biosynthesis contributed to protecting the cell shape and maintaining the bacterial growth of the psychrotolerant *B. cereus* group strain at the low temperature. The present results improve our understanding of the bacterial survival responses of mesophilic *B. cereus* group strains and psychrotolerant *B. cereus* group strains to low-temperature environments and should guide the development of strategies for controlling the growth of *B. cereus* groups, including psychrotolerant strains, at low temperatures.

## Figures and Tables

**Figure 1 microorganisms-09-01255-f001:**
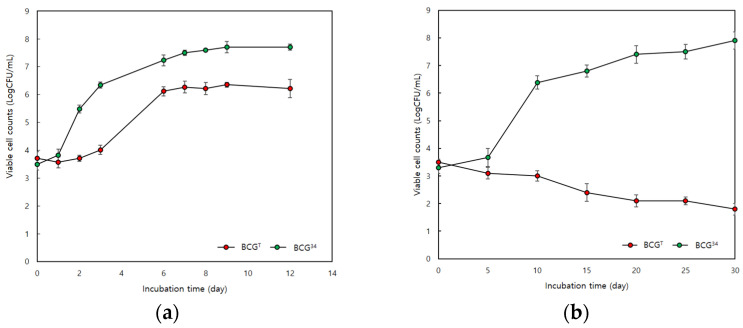
Growth curve of mesophilic *B. cereus* group strain (BCG^T^) and psychrotolerant *B. cereus* group strain (BCG^34^) at 10 °C (**a**) and 7 °C (**b**).

**Figure 2 microorganisms-09-01255-f002:**
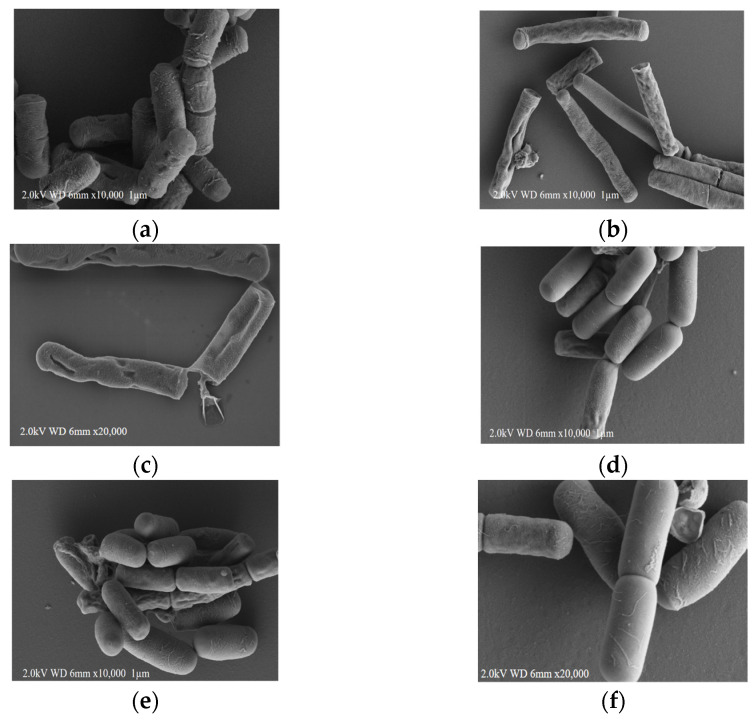
FE-SEM studies of mesophilic *B. cereus* group strain (BCG^T^) and psychrotolerant *B. cereus* group strain (BCG^34^) grown at 30 °C (**a**,**d**) and 10 °C (**b**,**c**,**e**,**f**). BCG^T^ at 30 °C (**a**), BCG^T^ at 10 °C (**b**,**c**), BCG^34^at 30 °C (**d**), and BCG^34^ at 10 °C (**e**,**f**).

**Figure 3 microorganisms-09-01255-f003:**
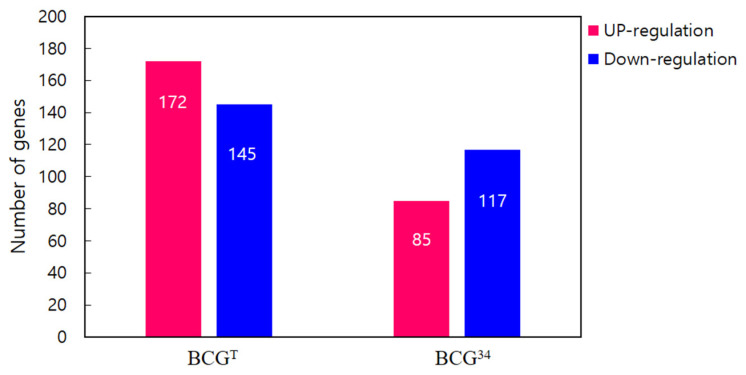
Differential gene expression of mesophilic *B. cereus* group strain (BCG^T^) and psychrotolerant *B. cereus* group strain (BCG^34^) grown at 10 °C compared to growth at 30 °C, respectively. Stacked bar graph representing the total number of the up- and downregulated genes identified in BCG^T^ and BCG^34^, considering a fold change greater than 2.0.

**Figure 4 microorganisms-09-01255-f004:**
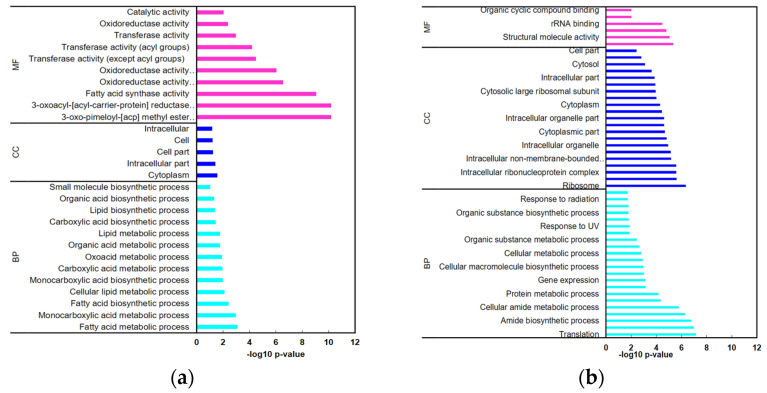
Functional annotation of the DEGs in the mesophilic *B. cereus* group strain (**a**) and the psychrotolerant *B. cereus* group strain (**b**) grown at 10 °C compared to growth at 30 °C. GO term analyses of transcriptomes categorized as Biological Process (BP), Cellular Component (CC), and Molecular Function (MF).

**Table 1 microorganisms-09-01255-t001:** Primer sequences for RT-qPCR.

Gene	Primer	Sequence (5′→3′)	Tm (°C)	GC (%)	Product Size (bp)
*uppP*	*uppP* -F	TCCAGCAGGTGTTATTGGTG	60.1	50.0	186
	*uppP* -R	GCTTGTGCTAATCCGACGAT	60.0	50.0	
*murG*	*murG* -F	GGTACAGCCGGACACGTTAT	59.9	50.0	155
	*murG* -R	CTCCTAAGCTTTCCCGTTGA	59.4	55.0	
*mraY*	*mraY* -F	AGCTTTAGGTGGAGCCATTG	59.3	60.0	210
	*mraY* -R	GTCACAACAACACGCCACTC	60.2	55.0	
*fabG*	*fabG* -F	GCAGCGAAAGTAGGACTTGT	57.2	60.0	182
	*fabG* -R	TCACCTGTTCCAGATCTACC	55.0	50.0	
*fabH*	*fabH* -F	GAATTAGGAGCAGACGGAAG	56.6	55.0	211
	*fabH* -R	ACCTCTCTCTTGCAGATTCC	56.0	50.0	
*fadE*	*fadE* -F	AAGAAGGAGAGCAAGCTAGG	55.7	50.0	152
	*fadE* -R	GCAATACCGTTACGACCTC	55.7	52.6	
*rpoB*	*rpoB*-F	CGGAGCTTGGTTAGAGTATG	55.2	50.0	230
	*rpoB*-R	CAGGACGTAGACGCTCATAA	56.6	50.0	

**Table 2 microorganisms-09-01255-t002:** Statistics of sequencing data assessment.

Name	Raw	Clean	Mapped	Uniquely Mapped	Splice	Gene
BCG^T^_C *	28,665,486	27,171,092 (94.8%)	26,397,864 (97.2%)	24,064,848 (88.6%)	758 (0.0%)	5015
BCG^T^_T	32,167,584	30,479,776 (94.8%)	29,567,793 (97.0%)	28,795,121 (94.5%)	2599 (0.0%)	5089
BCG^34^_C **	26,733,232	25,381,698 (94.9%)	23,359,519 (92.0%)	22,653,509 (89.3%)	4025 (0.0%)	4901
BCG^34^_T	36,039,900	34,424,698 (95.5%)	30,175,814 (87.7%)	14,568,814 (42.3%)	4051 (0.0%)	5047

* BCG^T^ _C, mesophilic *B. cereus* group ATCC 14579 activated at 30 °C; BCG^T^ _T, mesophilic *B. cereus* ATCC 14579 activated at 10 °C. ** BCG^34^_C, psychrotolerant *B. cereus* group isolated from foodstuff activated at 30 °C; BCG^34^_T, psychrotolerant *B. cereus* group activated at 10 °C.

**Table 3 microorganisms-09-01255-t003:** Differential gene expression related to lipid metabolism of the mesophilic *B. cereus* group strain (BCG^T^) and the psychrotolerant *B. cereus* group strain (BCG^34^) grown at 10 °C compared to growth at 30 °C.

Description	Gene ID	Gene	Product	Mesophilic *B. cereus* Group Strain (BCG^T^)		Psychrotolerant *B. cereus* Group Strain (BCG^34^)	
				Fold Change (Log_2_)	*p*-Value	Fold Change (Log_2_)	*p*-Value
Lipid metabolism	TBIG004756	*menE*	2-succinylbenzoate-CoA ligase	−2.79	0.037	2.97	0.219
	TBIG001316	*fabG*	3-oxoacyl-[acyl-carrier-protein] reductase	4.61	0.004	−0.57	0.044
	TBIG001751	*fabH2*	3-oxoacyl-[acyl-carrier-protein] synthase 3 protein 2	3.09	0.024	−0.83	0.039
	TBIG001966	*lcfA*	Long-chain-fatty-acid-CoA ligase	3.08	0.033	0.41	0.383
	TBIG005175	*fabZ*	3-hydroxyacyl-[acyl-carrier-protein] dehydratase	2.85	0.040	- *	-
	TBIG004904	*fadE*	Probable acyl-CoA dehydrogenase	4.63	0.039	3.05	0.027
	TBIG002725	*fadR*	Fatty acid metabolism regulator protein	6.84	0.016	0.75	0.272
Biotin metabolism	TBIG002459	*accC2*	Biotin carboxylase 2	5.15	0.025	−1.29	0.245
	TBIG002460	*yngHB*	Biotin/lipoyl attachment protein	4.64	0.013	−0.82	0.472

* Non-detection by RNA sequencing.

**Table 4 microorganisms-09-01255-t004:** Differential gene expression related to energy metabolism of mesophilic *B. cereus* group strain (BCG^T^) and psychrotolerant *B. cereus* group strain (BCG^34^) grown at 10 °C compared to growth at 30 °C.

Description	Gene ID	Gene	Product	Mesophilic *B. cereus* Group Strain (BCG^T^)		Psychrotolerant *B. cereus* Group Strain (BCG^34^)	
				Fold Change (Log_2_)	*p*-Value	Fold Change (Log_2_)	*p*-Value
Energy Metabolism	TBIG001419	*sat*	Sulfate adenylyltransferase	5.48	0.0001	−3.51	0.127
	TBIG001420	CYSC1_ BACHD	Probable adenylyl-sulfate kinase	5.68	0.0001	−3.09	0.184
	TBIG001418	*cysH*	Phosphoadenosine phosphosulfate reductase	4.12	0.003	−1.01	0.624
	TBIG002870	*ssuA*	Putative aliphatic sulfonates-binding protein	7.71	0.009	- *	-
	TBIG001323	*cysA2*	Sulfate/thiosulfate import ATP-binding protein	3.66	0.010	−1.76	0.432
	TBIG001098	*cysW*	Sulfate transport system permease protein	3.76	0.018	-	-
	TBIG001323	*cysA*	Sulfate/thiosulfate import ATP-binding protein	3.37	0.019	−1.76	0.432
	TBIG001324	*fbpB*	Ferric transport system permease protein	3.33	0.019	−1.27	0.558
	TBIG001097	*cysU*	Sulfate transport system permease protein	3.29	0.033	-	-
Cysteine metabolism	TBIG001873	*abrB*	Transition state regulatory protein	−2.85	0.047	−2.26	0.331
	TBIG003929	*metE*	5-methyltetrahydropteroyltriglutamate-homocysteine methyltransferase	3.82	0.035	−2.98	0.027
	TBIG005298	OAH *sulfhydrylase*	O-acetylhomoserine (thiol)-lyase	4.43	0.037	0.98	0.388
	TBIG002703	*yosT*	SPBc2 prophage-derived putative transcriptional regulator	4.44	0.032	-	-
Glutathione metabolism	TBIG002097	*bsaA*	Glutathione peroxidase homolog	3.16	0.020	1.67	0.469

* Non-detection by RNA sequencing.

**Table 5 microorganisms-09-01255-t005:** Differential gene expression related to amino acid metabolism of mesophilic *B. cereus* group strain (BCG^T^) and psychrotolerant *B. cereus* group strain (BCG^34^) grown at 10 °C compared to growth at 30 °C.

Description	Gene ID	Gene	Product	Mesophilic *B. cereus* Group Strain (BCG^T^)		Psychrotolerant *B. cereus* Group Strain (BCG^34^)	
				Fold Change (Log_2_)	*p*-Value	Fold Change Log_2_)	*p*-Value
Arginine metabolism	TBIG000429	*arcC*	Carbamate kinase	−7.04	0.0002	- *	-
	TBIG000427	*arcB*	Ornithine carbamoyltransferase	−8.65	0.001	-	-
	TBIG003492	*argH*	Argininosuccinate lyase	−6.94	0.012	0.99	0.560
	TBIG000426	*arcA*	Arginine deiminase	−6.09	0.012	-	-
	TBIG004050	*argF*	Ornithine carbamoyltransferase	−3.08	0.019	0.50	0.811
	TBIG004053	*argJ*	Arginine biosynthesis bifunctional protein	−2.95	0.044	4.00	0.048
	TBIG002004	*bltD*	Spermine/spermidine acetyltransferase	−3.51	0.035	0.77	0.734
	TBIG000205	*rocF*	Arginase	3.27	0.030	−1.58	0.031
BCAA metabolism	TBIG000890	*alsS*	Acetolactate synthase	−6.57	0.008	2.64	0.262
	TBIG001396	*ilvC1*	Ketol-acid reductoisomerase 1	−2.83	0.032	−3.31	0.177
	TBIG002461	*yngG*	Hydroxymethylglutaryl-CoA lyase	5.24	0.0003	−0.57	0.753
	TBIG001913	LIVB5_ BRUME	Leu/Ile/Val-binding protein homolog 5	3.23	0.016	5.31	0.047
Histidine metabolism	TBIG003587	*hutH*	Histidine ammonia-lyase	5.27	0.022	−2.61	0.226
	TBIG000514	*hutI*	Imidazolonepropionase	5.09	0.025	−2.69	0.254
Tryptophan metabolism	TBIG004527	Y4613_ BACCR	UPF0173 metal-dependent hydrolase	3.25	0.040		
	TBIG002724	*kynU*	Kynureninase	0.39	0.086	3.46	0.027
	TBIG002723	*kynB*	Kynurenine forma midase	0.30	0.096	3.06	0.021

* Non-detection by RNA sequencing.

**Table 6 microorganisms-09-01255-t006:** Differential gene expression related to carbohydrate metabolism of mesophilic *B. cereus* strain (BCG^T^) and psychrotolerant *B. cereus* strain (BCG^34^) grown at 10 °C compared to growth at 30 °C.

Description	Gene ID	Gene	Product	Mesophilic *B. cereus* Group Strain (BCG^T^)		Psychrotolerant *B. cereus* Group Strain (BCG^34^)	
				Fold Change (Log_2_)	*p*-Value	Fold Change (Log_2_)	*p*-Value
Glycolysis/ Gluconeogenesis	TBIG005105	*licH*	Probable 6-phospho-beta-glucosidase	−6.04	0.014	−1.42	0.533
	TBIG003111	*colA*	Colossin-A	−5.54	0.017	−0.96	0.655
	TBIG002269	*Hgd*	2-(hydroxymethyl) glutarate dehydrogenase	4.67	0.004	−0.41	0.822
	TBIG002742	*acoC*	Dihydrolipoyllysine-residue acetyltransferase component of acetoin cleaving system	−1.02	0.044	−5.99	0.026
	TBIG004774	*ldh2*	L-lactate dehydrogenase 2	−1.74	0.036	−6.63	0.012
	TBIG002741	*acoL*	Dihydrolipoyl dehydrogenase	−2.29	0.073	−6.51	0.017
	TBIG005034	*eno*	Enolase	−3.99	0.187	−4.05	0.049
	TBIG005229	*fba*	Fructose-bisphosphate aldolase	−1.76	0.041	−3.89	0.031
	TBIG003899	*pdhA*	Pyruvate dehydrogenase E1 component subunit alpha	−0.24	0.091	−3.86	0.044
	TBIG004898	*ldh3*	L-lactate dehydrogenase 3	0.37	0.076	−7.88	0.042
	TBIG005308	*cstA*	Carbon starvation protein A	1.95	0.160	4.29	0.018
	TBIG003044	*adhB*	Alcohol dehydrogenase 2	−1.26	0.298	4.39	0.044
	TBIG002744	*acoA*	2,6-dichlorophenolindophenol oxidoreductase subunit alpha	−0.70	0.058	−5.82	0.033
	TBIG003898	*pdhB*	Pyruvate dehydrogenase E1 component subunit beta	−0.70	0.755	−4.67	0.047
	TBIG002743	*acoB*	2,6-dichlorophenolindophenol oxidoreductase subunit beta	−1.79	0.052	−6.29	0.027
	TBIG001043	*glpD*	Aerobicglycerol−3-phosphate dehydrogenase	−0.83	0.516	−4.39	0.069
	TBIG000510	*pflA*	Pyruvate formate-lyase-activating enzyme	−5.44	0.141	−8.17	0.009
	TBIG002789	*ppaC*	Probable manganese-dependent inorganic pyrophosphatase	−0.65	0.600	−4.08	0.079
	TBIG001499	*gpsA*	Glycerol−3-phosphate dehydrogenase	−2.36	0.195	−3.91	0.085
Citrate cycle (TCA cycle)	TBIG001253	*odhA*	2-oxoglutarate ehydrogenase E1 component	−0.64	0.759	−4.08	0.048
	TBIG003896	*pdhD*	Dihydrolipoyl dehydrogenase	−0.84	0.684	−4.06	0.033
	TBIG001252	*odhB*	Dihydrolipoyllysine-residue succinyltransferase component of 2-oxoglutarate dehydrogenase complex	−0.84	0.704	−3.59	0.033
Butanoate metabolism	TBIG000891	*alsD*	Alpha-acetolactate decarboxylase	−6.96	0.007	1.78	0.428
	TBIG004083	*buk*	Probable butyrate kinase	−1.27	0.548	−3.38	0.045
Inositol phosphate metabolism	TBIG003692	PI-PLC	1-phosphatidylinositol phosphodiesterase	−3.69	0.008	−4.5	0.004
	TBIG002621	*Inos*	Inositol−3-phosphate synthase	3.49	0.027	- *	-
Pentose phosphate pathway	TBIG000359	*pcrB*	Heptaprenylglyceryl phosphate synthase	−3.18	0.029	0.79	0.970
	TBIG003617	*tkt*	Transketolase	−1.77	0.292	−4.96	0.031
	TBIG004012	*deoB*	Phosphopentomutase	−1.27	0.548	−3.38	0.045
Starch and sucrose metabolism	TBIG000435	*mapP*	Maltose 6′-phosphate phosphatase	−2.82	0.033	2.05	0.368
	TBIG003312	*ydzE*	Putative permease-like protein	3.53	0.014	0.98	0.993
Pyruvate metabolism	TBIG000509	*pflB*	Formate acetyltransferase	−3.97	0.039	−8.65	0.251
	TBIG002463	*yngE*	Uncharacterized carboxylase	4.49	0.042	−1.84	0.381
Propanoate metabolism	TBIG002267	*prpB*	Methylisocitrate lyase	4.34	0.003	−2.18	0.244
Glyoxylate and dicarboxylate metabolism	TBIG002265	*mmgD*	2-methylcitrate synthase	5.32	0.024	−1.07	0.524
	TBIG003865	*amiF*	Formamidase	3.04	0.033	2.28	0.240
Amino sugar and nucleotide sugar metabolism	TBIG003557	*nodB*	Chitooligosaccharide deacetylase	2.72	0.038	3.49	0.200

* Non-detection by RNA sequencing.

**Table 7 microorganisms-09-01255-t007:** Differential gene expression related to the cell membrane and cell wall of the mesophilic *B. cereus* group strain (BCG^T^) and the psychrotolerant *B. cereus* group strain (BCG^34^) grown at 10 °C compared to growth at 30 °C.

Description	Gene ID	Gene	Product	Mesophilic *B. cereus* Group Strain (BCG^T^)		Psychrotolerant *B. cereus* Group Strain (BCG^34^)	
				Fold Change (Log_2_)	*p*-Value	Fold Change (Log_2_)	*p*-Value
Peptidoglycan biosynthesis	TBIG003837	*murG*	UDP-N-acetylglucosamine--N-acetylmuramyl-(pentapeptide) pyrophosphoryl-undecaprenol N-acetylglucosamine transferase	−6.79	0.012	2.21	0.033
	TBIG000690	*uppP*	Undecaprenyl-diphosphatase	−5.03	0.018	1.27	0.047
	TBIG003840	*mraR*	Phospho-N-acetylmuramoyl- pentapeptide-transferase	−1.81	0.047	1.76	0.030
	TBIG005169	*epsC*	UDP-N-acetylglucosamine 4,6-dehydratase	−3.14	0.033	2.67	0.036
Cell growth factor	TBIG002125	*spoVS*	Stage V sporulation protein S	2.97	0.046	3.05	0.191
	TBIG002015	*cotH*	Inner spore coat protein H	2.63	0.042	2.79	0.241
	TBIG001634	*flgB*	Flagellar basal body rod protein	3.57	0.030	- *	-

* Non-detection by RNA sequencing.

## Data Availability

The NGS data presented in this study are available in the NCBI GenBank under the accession numbers CP034551 and PRJNA727406.

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
