# Peer review of "Morphological Features and Cold-Response Gene Expression in Mesophilic Bacillus cereus Group and Psychrotolerant Bacillus cereus Group under Low Temperature"

_microorganisms, 2021, doi:10.3390/microorganisms9061255_

Round 1
Reviewer 1 Report
This manuscript is generally well written and structured. I am strongly inclined to recommend it for publication in Microorganisms, although I have some questions that should be addressed in modifications to the text, which I include here below.
1) Paragraph 2.4: you explain that you used rpoA gene as the reference gene to normalize and estimate up or down-regulation of the selected genes but in the table 1 you reported the sequences of rpoB and immediately after you said again rpoA. Please correct the wrong version.
2) In the legen of figure 2 (line 204) I suggest to introduce some details in parentheses as reported:
“Figure 2. FE-SEM studies of mesophilic (BCG T ) and psychrotolerant (BCG 34 ) B. cereus group isolates grown at 30°C (a and c) and 10°C (b and d)….
3) in the paragraph 3.2. “Microscopic Observations of Mesophilic and Psychrotolerant B. cereus Group Strains at Low Temperature”. In the lines 213-215 you said that some cell walls and
membranes of BCG T were partially disrupted. I think that is not so evident from the FE-SEM images and in particular is not indicative of the % of cells disrupted. Did you try to conduct the very easy experiment using Dapi and PI staining?
4) line 245: 175 genes you have to change with 172 genes.
5) there is a problem with the figure 4 (a) and (b) are not indicative of mesophilic and psychrotolerent B.cereus groups. Please put in the correct way.
Author Response
Reviewer 1
This manuscript is generally well written and structured. I am strongly inclined to recommend it for publication in Microorganisms, although I have some questions that should be addressed in modifications to the text, which I include here below.
1) Paragraph 2.4: you explain that you used rpoA gene as the reference gene to normalize and estimate up or down-regulation of the selected genes but in the table 1 you reported the sequences of rpoB and immediately after you said again rpoA. Please correct the wrong version.
I revised reference gene name to ‘rpoB’ from ‘rpoA’ (Materials and Methods section 2.5).
2) In the legend of figure 2 (line 204) I suggest to introduce some details in parentheses as reported:
“Figure 2. FE-SEM studies of mesophilic (BCG T ) and psychrotolerant (BCG 34 ) B. cereus group isolates grown at 30°C (a and c) and 10°C (b and d)….
I revised the title of figure 2 according to your comments (Figure 2. Line 231).
3) in the paragraph 3.2. “Microscopic Observations of Mesophilic and Psychrotolerant B. cereus Group Strains at Low Temperature”. In the lines 213-215 you said that some cell walls and membranes of BCG T were partially disrupted. I think that is not so evident from the FE-SEM images and in particular is not indicative of the % of cells disrupted. Did you try to conduct the very easy experiment using Dapi and PI staining?
Thank you for your comment. In this study, we would like to confirm the morphological change of the bacterial cell wall under low temperature. To cultivate the viable vegetative cell for analysis of RNA extraction, growth capability, and FE-SEM in two type of B. cereus group strains, mesophilic B. cereus group strain and psychrotolerant B. cereus group strain was incubated overnight at 30°C. The overnight cultures of BCGT and BCG34 activated at 10°C diluted to a final concentration of 1 x 103 log CFU/mL and then incubated in triplicate at 10°C for 12 days to confirm the exponential growth phage (growing cells) of vegetative cells. The bacterial suspension cultured at time intervals (0, 2, 4, 6, 8, 10, 12 days for 10°C) and the cell were counted to determine bacterial cell count (CFU/mL) at each time point. Although the cells were harvested in viable vegetative cell status, some cell showed disrupted shape in FE-SEM. Previous figure was not enough to show the changed cell shape, thus we inserted the new image for FE-SEM results. In present study, we did not consider the cell viability assay by DAPI staining as the cells were harvested at the exponential growth phage. However, our further study will consider the cell viability assay for evaluating of disrupted cells using DAPI and PI staining.
4) line 245: 175 genes you have to change with 172 genes.
I revised the gene number to 172 (Line 245).
5) there is a problem with the figure 4 (a) and (b) are not indicative of mesophilic and psychrotolerent B.cereus groups. Please put in the correct way.
I corrected the figure position.

Reviewer 2 Report
The manuscript is dedicated to the study of cold-resistant strain of Bacillus cereus and provide transcriptomic data for the reaction of mesophilic and psychrotolerant strains to cold-stress.
The general remarks:
- The work seems to be too descriptive. No insights into mechanism of cold-resistance are provided. It is obvious that the reaction to cold stress should be different for the two strains, but what helps BCG34 to overcome it?
- The work consists of two parts: description of EM of the Bc strains and transcriptomic study. But they seems to be too independent. The phenotypic changes are not explained by transcriptomes.
More detailed concerns:
- The process of BCGt adaptation may interfere with the experiment. I think, the BCG34 strain should undergo the same procedure.
- Why rRNA at line 117?
- As far as I understand only one sample of BCGt and BCG34 in each growth condition has been sequenced. It is not enough to make any statistically significant conclusions about DEGs.
- What is BCG reference genome (the authors have provided five accessions)? Which one has been used? Considering great variability of BCG strains, it is better to use the closest genome possible.
- What program has been used for pathway enrichment analysis?
- Figure 1 - no 7C growth condition has been mentioned in M&M section
- I would move section 3.4 to M&M, it adds nothing to the results of the study.
Author Response
Reviewer 2
The manuscript is dedicated to the study of cold-resistant strain of Bacillus cereus and provide transcriptomic data for the reaction of mesophilic and psychrotolerant strains to cold-stress.
The general remarks:
1.The work seems to be too descriptive. No insights into mechanism of cold-resistance are provided. It is obvious that the reaction to cold stress should be different for the two strains, but what helps BCG34 to overcome it?
The Bacillus cereus group comprises more than 20 species with close genetic similarity, and the species are difficult to distinguish into different species according to 16S rRNA gene sequences. Nine species among these groups (B. anthracis, B. cereus, B. cytotoxicus, B. mycoides, B. pseudomycoides, B. thuringiensis, B. toyonensis, B. weihenstephanensis, and B. wiedmannii) can cause anthrax or foodborne illness in humans or insects. Bacillus cereus group is not traditionally considered a psychrotrophilic species, but some B. cereus groups (psychrotolerant B. cereus group) can grow at below 10°C without adaptation processing unlike common B. cereus group. In a previous study, we isolated a psychrotolerant species from fresh vegetables distributed by cold chain and showed a growth capability at less than 10°C. Furthermore, when activating at low temperature, some among these isolates showed greater tolerant response to environmental stress factors than those at optimal temperature. The consumer demand for refrigerated foods has increased due to the retention of nutritional and sensorial qualities in foodstuffs. However, cold storage might provide an environment that is favorable for bacterial survival and the growth of psychrotolerant strains. Previous study about bacterial response to low temperature were limited to a small number of characteristics, providing not enough information about comprehensive understating of bacterial stress response. Furthermore, limited information is available about the impact of low temperature on the bacterial growth of psychrotolerant B. cereus group. In this study, we investigated whether psychrotolerant B. cereus group show a similar response to grow under low temperature with mesophilic B. cereus group. According to our results, mesophilic B. cereus group strain and psychrotolerant B. cereus group strain utilize different strategies for survival under cold conditions. The mesophilic B. cereus group strain use common bacterial stress mechanisms to grow under low temperature. For example, this strain exhibited increased sulfur assimilation, induction of cysteine and glutathione biosynthesis, and upregulation of fatty acid biosynthesis for bacterial growth under cold conditions. Whereas, carbohydrate metabolism in psychrotolerant B. cereus group strain was repressed to save energy for growth, while upregulation of genes responsible for the catabolism of typtophan and alcohols was modulated to energy production by providing alternate pathway for NAD regeneration and balance of ATP level. Furthermore, increased peptidoglycan biosynthesis contributed to protect cell shape and maintain the bacterial growth of psychrotolerant B. cereus group strain under low temperature. Our study will provide a new insight about genomic, transcriptomic, and morphological characteristics of the psychrotolerant B. cereus group isolates from information and a scientific information for improving bacterial inhibition strategies in the food industry.
The contents of mentioned above inserted in Introduction and Conclusion of revised manuscript.
2.The work consists of two parts: description of EM of the Bc strains and transcriptomic study. But they seems to be too independent. The phenotypic changes are not explained by transcriptomes.
Most of bacteria possess a peptidoglycan cell wall to withstand environmental changes and to maintain rod cell shape. Environmental stress prevents peptidoglycan formation and remodeling during cell division throughout inhibition of peptidoglycan biosynthesis (Waxman et al., 1983). Environmental fluctuations may challenge the peptidoglycan biosynthesis mechanisms by disrupting enzymatic activity or regulatory protein-protein interactions. The shape of bacterial cell is maintained by the peptidoglycan layer that encases the cytoplasmic membrane to protect the cell. The enzymes (N-acetylmuramidases, N-acetylglucosaminidases, amidases, endopeptidases and carboxypeptidases) related to peptidoglycan have major roles in peptidoglycan turnover during growth and cell division, cell shape maintenance or bacterial interaction (Vollmer et al., 2008). Proteins encoded by the mra and mur genes maintain and control the bacterial membrane (Laddomada et al., 2016). The shape-determining structure in bacterial cell membrane is made up of peptidoglycan, the main component of the cell wall. In rod shape cell, mur genes showed a significant upregulation under stress conditions (Cava et al., 2014). In psychrotolerant B. cereus group strain, the genes related to peptidoglycan and cell wall biosynthesis were increased at low temperature compared with the levels found at the optimal temperature; specifically, 1.2- to 2.7-fold upregulation of three genes, murG1 (2.21 log2FC), uppP1 (1.27 log2FC), and epsC (2.67 log2FC), was observed in RNA-seq. In addition, the results of RT-qPCR were consistent with RNA-seq data. Low temperature resulted in significantly upregulated expression of uppP (1.9-fold), murG (3.8-fold), and mraY (3.0-fold) genes involved in peptidoglycan biosynthesis in psychrotolerant B. cereus group strain. Upshift of peptidoglycan biosynthesis might contribute to response to maintain the growth capability and cell shape as shown in the expression pattern of uppP, murG, and mraY genes as well as unchanged cell morphology under low temperature compared to optimal temperature.
More detailed concerns:
1.The process of BCGt adaptation may interfere with the experiment. I think, the BCG34 strain should undergo the same procedure.
As mentioned above, Bacillus cereus group is not traditionally considered a psychrotrophilic species, but some B. cereus groups (psychrotolerant B. cereus group) can grow at below 10°C without adaptation processing unlike common B. cereus group. The adaptation response of B. cereus to cold stress has been studied, but limited information is available about the impact of low temperature on the growth of psychrotolerant B. cereus group strains. In particular, no studies have compared the two types of B. cereus groups (mesophilic or psychrotolerant species) to evaluate their different or varied growth capabilities under low temperature. In food industry, mesophilic B. cereus also can grow at low temperature throughout the prolonged cold exposure (cold-adaptation) during refrigerated storage or cold distribution thus both mesophilic B. cereus group and psychrotolerant B. cereus group can observe in foodstuffs. To develop the traditional strategies for bacterial inhibition, we compared the two type of B. cereus group to evaluate their different or varied growth capabilities under low temperature. We collected B. cereus ATCC 14579 as mesophilic strain because this strain was used as reference (type) strain of common B. cereus. The strain for psychrotolerant B. cereus group collected BCG-34 shown high grow rate and stress tolerance response at low temperature in previous our study (Park et al., 2020). As B. cereus ATCC 14579 did not grow at 10°C, we gradually exposed this strain to low temperature for the adaptation of mesophilic B. cereus group strain to low temperature. Whereas, psychrotolerant B. cereus group strain that is capable to grow at low temperature was activated at 30°C, transferred to 10°C and incubated for 10 days (OD600>0.6) without cold adaptation processing.
2.Why rRNA at line 117?
According to your comments, I need to make changes to the sentence I wrote in materials and methods. I correctly revised the sentence.
3.As far as I understand only one sample of BCGt and BCG34 in each growth condition has been sequenced. It is not enough to make any statistically significant conclusions about DEGs.
The analysis for growth capability and RT-qPCR were carried out in triplicate and duplicates of extracted RNA samples were used for RNA-sequencing.
4.What is BCG reference genome (the authors have provided five accessions)? Which one has been used? Considering great variability of BCG strains, it is better to use the closest genome possible.
The Bacillus cereus group comprises more than 20 species with close genetic similarity, and the species are difficult to distinguish into different species according to 16S rRNA gene sequences. We could not correctly identify food isolate (BCG34) by 16S rRNA sequencing and this isolate matched into five reference strain among B. cereus group to obtain high-quality clean reads for RNA-seq. Finally, reads from BCG34 were mapped to the complete sequenced genomes of NZ_CP000903. I inserted the mapped accession number for psychrotolerant B. cereus group in Materials and Methods section 2.4.2.
5.What program has been used for pathway enrichment analysis?
In this study, David 6.8 was used to perform the enrichment analysis of RNA-seq data. I revised the Method and Material.
6.Figure 1 - no 7C growth condition has been mentioned in M&M section
The method for bacterial growth capability at low temperature (10°C and 7°C) were inserted in Materials and Methods section 2.2.
7. I would move section 3.4 to M&M, it adds nothing to the results of the study.
Thank you for your comment. In this study, the expression profiles of selected genes encoding peptidoglycan biosynthesis and fatty acid biosynthesis by RNA-seq were analyzed by RT-qPCR to investigate and validate the effects induced by low temperature in mesophilic B. cereus group strain and psychrotolerant B. cereus group strain. In our results, the fabG (3.7-fold), fabH (3.0-fold) and fadE(4.6-fold) gene encoding fatty acid biosynthesis showed increased expression in BCGT and the decreased expression of fabG (0.05) and fabH (-0.9) gene observed in BCG34. Low temperature resulted in significantly upregulated expression of uppP (1.9-fold), murG (3.8-fold), and mraY (3.0-fold) genes involved in peptidoglycan biosynthesis in BCG34. Whereas the expression of these genes (uppP, -1.2-fold; murG, -2.8-fold; mraY, -0.4-fold) down-regulated in BCGT at 10°C compared to at 30°C. Unlike mesophilic strain, fatty acid biosynthesis may be important for bacterial growth under low temperature in psychrotolerant strain. However, upshift of peptidoglycan biosynthesis might contribute to response to maintain the growth capability and cell shape as shown in the expression pattern of uppP, murG, and mraY genes as well as unchanged cell morphology under low temperature compared to optimal temperature. The results of mentioned above were inserted in Results and Discussion section 3.4.

Round 2
Reviewer 2 Report
I would like to thank the authors for addressing my comments. I keep the numbers of my previous comments for consistency.
Major comments:
2. Thank you for the explanation. But still I would draw more attention to the connection between morphology and transcriptomic data in the manuscript to improve the readability.
Minor comments:
- I do understand why BCGt undergo the adaptation process. But I think the transcriptoms of BCG34 strains with adaptation stage and without it should be compared.
- .
- Usually, it is quite difficult work with RNA-seq data made in duplicates. I mean, it is difficult to prove statistical significance, or you may have big false negative rate. I would recommend to do RNA-seq also in triplicates at least.
- .
- Now David is mention in the manuscript as a tool for GO term enrichment analysis. Tool for KEGG pathway enrichment is still unknown.
- .
- Still I do not understand. Have you found new effects (changes in gene expression) using qPCR, which you were unable to detect with RNA-seq experiments? If no new effects have been found, it is just validation and should be moved to M&M section.
Author Response
Thank you for providing these insights about our manuscript one more time.
I will attempt to answer you sincerely.
1. I do understand why BCGt undergo the adaptation process. But I think the transcriptoms of BCG34 strains with adaptation stage and without it should be compared.
Thanks for bringing this matter to our attention. As it was previously mentioned, psychrotolerant BCG34 was able to grow at low temperature without cold adaptation stage unlike mesophilic BCG. There are few studies about morphology, enterotoxin gene profile, antibiotic susceptibility, stress tolerance, and growth mechanisms of psychrotolerant B. cereus group, especially food isolates. Thus, we thought that it is important to understand the growth characteristics of psychrotolerant B. cereus group isolates (which is able to grow at low temperature without cold adaptation) to improve the food safety during cold chain. If psychrotolerant BCG34 undergo the cold-adaptation process like mesophilic B. cereus group strain, this strain adapted at low temperature may display different characteristics such as cross-protection compared with the strain without cold adaptation stage (Guerin et al., 2016; Dargaignaratz et al., 2016; Choma et al., 2000). Since we investigated whether psychrotolerant B. cereus group share a similar response to grow at low temperature with cold-adapted mesophilic B. cereus group, BCG34 did not undergo cold adaptation stage in present study. However, our further study will consider the cold adaptation process for these strains to improve our understanding about psychrotolerant B. cereus group.
2. Usually, it is quite difficult work with RNA-seq data made in duplicates. I mean, it is difficult to prove statistical significance, or you may have big false negative rate. I would recommend to do RNA-seq also in triplicates at least.
Thanks for your comments. We primarily confirmed the morphology to compare the cold response between mesophilic B. cereus group and psychrotolerant B. cereus group. According to the results of SEM, some bacterial cells of mesophilic B. cereus group strain showed disrupted or elongated cell shape, whereas cell shape of psychrotolerant B. cereus group strain retained under low temperature. To understand the morphology of mesophilic B. cereus group strain and psychrotolerant B. cereus group strain, the transcriptomes of these strains were analyzed using RNA-seq. Transcriptome analysis indicated that the expression of some genes involved in cell wall modification, such as peptidoglycan biosynthesis showed different pattern between two types of B. cereus group under cold condition. And the results of RNA-qPCR were similar with RNA-seq results, indicating that the RNA-seq analysis may be reliable. Although the present study was work with RNA-seq made in duplicates, the results of SEM, RNA-seq and RNA-qPCR may help understand the survival strategies of mesophilic B. cereus group strain and psychrotolerant B. cereus group strain at low temperature. However, we will try so many different ways such as gene expression, protein expression, and metabolite profiling to provide a significant results of transcriptomic analysis in further study according to your comments.
3. Now David is mention in the manuscript as a tool for GO term enrichment analysis. Tool for KEGG pathway enrichment is still unknown.
In this study, the GO enrichment analysis and KEGG pathway enrichment analysis were performed using DAVID bioinformatics resources 6.8. This text was inserted in Material and Method section 2.4. (Line 161)
4. Still I do not understand. Have you found new effects (changes in gene expression) using qPCR, which you were unable to detect with RNA-seq experiments? If no new effects have been found, it is just validation and should be moved to M&M section
I accepted your comments. The text related to validation by RNA-qPCR was deleted in Results and Discussions section 3.4 and the sentence ‘The results of RNA-qPCR was consistent with RNA-seq data’ was moved in Methods and Materials (Line 192).